# Using social choice theory and acceptability analysis to measure the value of health systems

Hai Shen[1], Yubing Sui[2]*, Yelin Fu[3]

**1** Business School, Xi'an International Studies University, Xi'an, China, **2** College of Economics, Shenzhen University, Shenzhen, China, **3** Department of Industrial & Manufacturing Systems Engineering, The University of Hong Kong, Hong Kong SAR

\* suiyubing@szu.edu.cn

## Abstract

The Future Health Index (FHI) is developed by the Royal Philips to help determine the readiness of countries to address global health challenges and build sustainable, fit-for-purpose national health systems. The FHI 2018 presents the Value Measure to measure the value of 16 health systems, which is formulated by taking the arithmetic average of Access, Satisfaction and Efficiency. However, this scheme is not the Pareto optimal and loses association with weights. For these reasons, this paper proposes to apply the social choice theory and Stochastic Multicriteria Acceptability Analysis for group decision making (SMAA-2) to measure the value of health systems, by means of re-constructing the Value Measure. Specifically, we begin with considering all possible individual preferences among Access, Satisfaction and Efficiency, which is mathematically represented by ranked weights of them; the pessimistic and optimistic outcomes under certain individual preference are derived in a closed-form manner, according to which an interval decision matrix is then formulated; the SMAA-2 is then lastly applied to compute the holistic acceptability index, which is considered as a revised Value Measure. An empirical study using the data of 16 health systems is conducted to show the effectiveness and superiority of our method. It is demonstrated that our method always outperforms the Value Measure, by means of comparing the Spearman's rank correlation coefficients.

## Introduction

*The challenges of delivering health care in many countries are receiving increasing attentions as costs continue to rise and evidence of uneven quality accumulates* [1]. Although most health care reforms have focused on coverage, the far bigger long-term driver of success will originate from restructuring the health care delivering system to a value-based system [2]. The concept of value-based health care suggests a change of model in which the provision of health services does not focus on the quantity of services provided but on the value they generate, understanding value as overall quality of care and health outcomes related to the costs achieving those

**Funding:** This research is financially supported by Humanities and Social Science Research Projects

of Ministry of Education of the People's Republic of China (Funding No.18YJC790142), Humanities and Social Science Research Projects of Guangdong Province (Funding No. GD18CGL06), Humanities and Social Science Research Projects of Shenzhen (Funding No. SZ2018C012).

**Competing interests:** The authors have declared that no competing interests exist.

outcomes. In this sense, as a people-centric approach, value-based health care describes a system with the goal of increasing access to care, improving patient outcomes, and delivering satisfaction to both patients and practitioners at optimum cost. In other words, value-based health care is contextual, geared towards providing the right care in the right place, at the right time and at the right level of cost. Therefore, achieving high value for various stakeholders must become the overwhelming goal of health care delivery. Rigorous, disciplined measurement and improvement of value are the best way to drive system progress [3]. Nevertheless, the value in health remains largely misunderstood and unmeasured.

*The Future Health Index (FHI) is a research-based platform designed by the Royal Philips to help determine the readiness of countries to address global health challenges and build sustainable, fit-for-purpose national health systems.* The FHI 2016 measures perceptions to produce a snapshot of how health care is experienced on both sides of the patient-professional divide. The FHI 2017 compares these perceptions to the reality of health systems in each country researched. The FHI 2018 builds on the increasing consensus that, with the rise of chronic diseases and health care costs, the value-based care model is the best approach to address these challenges. In addition, the FHI 2018 identifies key challenges that form a barrier to the large-scale adoption of value-based care and improved population access; and assesses where connected care technology—data collection and analytics, and telehealth—can help speed up the health care transformation process. The FHI 2018 measures and assesses the value presented in 16 health systems of developed and developing markets through proposing a broadly applicable composite indicator, namely, Value Measure. The Value Measure combines criteria with respect to value-based health care and access to care, arguably the ultimate goals of modern health care.

The Value Measure consists of three metrics: *Access* (how universal, and affordable, is access to health care in the designated market?), *Satisfaction* (to what extent do the general population and practitioners in the designated market see the health care system as trustworthy, and effective?) and *Efficiency* (does the system in the given market produce outcomes at an optimum cost?). The components of Value Measure are listed in Table 1. Each metric is composed of several sub-metrics, which are normalized to ensure comparability across countries and are scored to fit onto a 0 to 100 scale. The scores for each sub-metric are arithmetically averaged to calculate each metric sore and those scores are then arithmetically averaged to construct the Value Measure. That is,

$$\text{Value Measure} = \frac{\text{Access} + \text{Satisfaction} + \text{Efficiency}}{3}. \tag{1}$$

**Table 1. Value measure.**

| Value Measure | Access | Skilled health professional density (per 10,000 population) |
|---|---|---|
| | | Risk of impoverishment due to surgical care (% of people at risk) |
| | | Hospital beds (per 10,000 population) |
| | Satisfaction | Trust in health care system (HCPs and general population) |
| | | Health care system meets needs (HCPs and general population) |
| | | Rating of health care system overall (HCPs) |
| | Efficiency | Health care spend as a percentage of GDP |
| | | Tuberculosis: incidence and treatment success rates |
| | | Life and healthy life expectancy at birth |
| | | Probability of dying from key chronic diseases between 30 and 70 |
| | | Neonatal mortality rate |
| | | Maternal mortality rate |

The scores for these sub-metrics use a combination of third-party data and survey data. Specifically, the third-party data is sourced from many organizations including the World Health Organization, The Commonwealth Fund, and the World Bank, while the survey data is collected from the countries analyzed using their native language. A combination of face-to-face, online and phone interviewing is employed. The sample from the survey includes 24,654 adults and 3,244 health care professionals.

As shown in (1), the Value Measure assigns equal weights to Access, Satisfaction and Efficiency, this plausible scheme results in substantial information loss [4]. In addition to this, the arithmetical average is significantly affected by the extreme values, not Pareto optimal, and losses association with weights [5]. All these shortcomings inspire scholars and practitioners to develop new methods for improving the calculation of the Value Measure. The contribution of this paper is the development of a new method to modify the Value Measure released by the Royal Philips for measuring the value of health systems, based upon the social choice theory and Stochastic Multicriteria Acceptability Analysis for group decision making (SMAA-2). Social choice is the theory of how one designs or chooses a mechanism to summarize from a set of individual preference orders over alternatives available to a society of those individuals to a collective or social preference order over those same alternatives [6, 7]. Stochastic Multi-criteria Acceptability Analysis (SMAA) is a multicriteria decision support method for multiple experts in discrete problems, based on exploring the weight space to describe the valuations that make each alternative the preferred one [8, 9]. SMAA-2 extends SMAA by taking into account information about other ranking positions, therefore identifies good compromise alternatives.

Specifically, we begin with eliminating the equalitarianism assumption to consider all possible individual preferences among three metrics. Certain individual preference is mathematically represented by a set of ranked weights. It seems reasonable that a decision maker should at least rank the metrics, since rankings are normally easier to provide than usually inaccessible precise weights information [4, 10]. In the meanwhile, the decision maker may be unable, unavailable, or even unwilling to obtain sufficiently precise weights [11]. Nevertheless, it is difficult to achieve consensus about exact weights in a problem with multiple decision makers [9]. In this sense, we then calculate the worst and best outcomes under certain individual preference in a closed-form manner, according to which an interval-valued decision matrix is formulated with country-as-row and individual preference-as-column. Lastly, the SMAA-2 is applied to obtain the holistic acceptability index for each country, which is regarded as an improved version of the Value Measure. We compute the Spearman's rank correlation coefficients to demonstrate the superiority and rationality of the proposed method. This study proposes a new incentive and a feasible direction to measure the value of health systems in an appropriate manner, along with the provision of some academic, managerial and policy-related implications.

The remainder of the paper is organized as follows. We present the method for improving the evaluation of the Value Measure in Section 2, followed by an empirical study for a panel of 16 countries in Section 3. We conclude in Section 4 by discussing the details of our method and suggestions for future research.

## 2. Method

For the purpose of measuring the value of health systems that are previously aggregated using the arithmetic average, this section proposes a method for the general case with $m$ Decision Making Units (DMUs) and $n$ metrics, which can be easily applied to improve the Value Measure with Access, Satisfaction and Efficiency. $x_{ij}$, $i = 1, 2, \ldots, m$, $j = 1, 2, \ldots, n$ indicates the

performance of DMU $i$ under sub-index $j$. To adjust values measured on different scales to a notionally common scale, we use the feature scaling (alternative known as min-max normalization) to scale the range in [0, 1]:

$$z_{ij} = \frac{x_{ij} - \min_i \{x_{ij}\}}{\max_i \{x_{ij}\} - \min_i \{x_{ij}\}}, i = 1, 2, \ldots, m, j = 1, 2, \ldots, n.$$

The method proposed is two-fold and begins with investigating all possible individual preferences among the $n$ metrics, under which the pessimistic and optimistic outcomes are derived in a closed-form manner; we then employ the SMAA-2 to compute the holistic acceptability index for aggregating the individual preferences into a social choice result.

## 2.1 Individual preference

This paper takes into account all possible individual preferences among the metrics to deal with the drawbacks associated with the arithmetic average method. In this sense, an individual preference can be represented by an importance order of metrics. For the ease of demonstration, we only investigate one of the individual preferences in this section, the result of which can be easily migrated in other scenarios. We investigate the situation in which $w_1 \geq w_2 \geq \cdots \geq w_n$, and $w_j, j = 1, 2, \ldots, n$ is the importance degree of metric $j$. In this manner, the pessimistic and optimistic results for DMU $i$ can be determined by the following two linear programs:

$$v_i^p = \min \sum_{j=1}^n z_{ij} w_j$$
$$s.t. \quad w_1 \geq w_2 \geq \cdots \geq w_n \tag{2}$$
$$\sum_{j=1}^n w_j = 1, w_j \geq 0.$$

$$v_i^o = \max \sum_{j=1}^n z_{ij} w_j$$
$$s.t. \quad w_1 \geq w_2 \geq \cdots \geq w_n \tag{3}$$
$$\sum_{j=1}^n w_j = 1, w_j \geq 0.$$

For $\alpha_j \geq 0, j = 1, 2, \ldots, n$, we define the weights as $w_k = \sum_{j=k}^n \alpha_j$. This is consistent with given individual preference among metrics, $w_1 \geq w_2 \geq \cdots \geq w_n$. Let $\beta_j = j\alpha_j$,

$$\sum_{j=1}^n \beta_j = \sum_{j=1}^n j\alpha_j = \sum_{j=1}^m \left( \sum_{k=j}^n \alpha_k \right) = \sum_{j=1}^n w_j = 1. \tag{4}$$

Moreover, we define $s_{ik} = \frac{1}{k} \sum_{j=1}^k z_{ij}, k = 1, 2, \ldots, n$, then

$$\sum_{j=1}^n z_{ij} w_j = \sum_{j=1}^n \sum_{k=j}^n z_{ij} \alpha_k = \sum_{j=1}^n \sum_{k=j}^n z_{ij} \left( \frac{1}{k} \beta_k \right) = \sum_{k=1}^n \beta_k \left( \frac{1}{k} \sum_{j=1}^k z_{ij} \right) = \sum_{k=1}^n \beta_k s_{ik}. \tag{5}$$

Therefore, the linear program (3) is equivalent to the following model:

$$v_i^o = \max \sum_{k=1}^{n} \beta_k s_{ik}$$

$$s.t. \sum_{k=1}^{n} \beta_k = 1, \beta_k \geq 0.$$

(6)

Let $\hat{k} \in \{1, 2, \ldots, n\}$ satisfies that $s_{i\hat{k}} = \max_k \{s_{ik}\}$, then the optimal solution to linear program (6) is determined by

$$\beta_k = \begin{cases} 1, & k = \hat{k}; \\ 0, & \text{otherwise.} \end{cases}$$

(7)

Consequently, the optimistic result for DMU $i$ with certain individual preference can be easily determined as the following closed form: $v_i^o = \max_k \{s_{ik}\} = \max_k \left\{ \frac{1}{k} \sum_{j=1}^{k} z_{ij} \right\}, k = 1, 2, \ldots, n.$

This scheme is easy-to-understand and simple-to-implement, and can be readily migrated to other situations. Similarly, the pessimistic result for DMU $i$ with certain individual preference can be derived as $v_i^p = \min_k \{s_{ik}\} = \min_k \left\{ \frac{1}{k} \sum_{j=1}^{k} z_{ij} \right\}, k = 1, 2, \ldots, n.$

Taking into account the pessimistic and optimistic outcomes under all possible individual preferences, an interval-valued decision matrix $\Omega_{m \times n!}$ is formulated as below:

$$\Omega_{m \times n!} = \begin{bmatrix} [v_{11}^p, v_{11}^o] & \cdots & [v_{1t}^p, v_{1t}^o] & \cdots & [v_{1n!}^p, v_{1n!}^o] \\ [v_{21}^p, v_{21}^o] & \cdots & [v_{2t}^p, v_{2t}^o] & \cdots & [v_{2n!}^p, v_{2n!}^o] \\ \vdots & \vdots & \vdots & \vdots & \vdots \\ [v_{m1}^p, v_{m1}^o] & \cdots & [v_{mt}^p, v_{mt}^o] & \cdots & [v_{mn!}^p, v_{mn!}^o] \end{bmatrix}$$

(8)

As claimed by [12], $\Omega_{m \times n!}$ represents a stochastic decision problem. SMAA-2 has been accepted as an effective tool to solve this problem [9].

## 2.2 SMAA-2

Stochastic multicriteria acceptability analysis (SMAA) is a multicriteria decision support method for multiple experts in discrete problems, based on exploring the weight space to describe the valuations that make each alternative the preferred one [8, 9]. SMAA-2 extends SMAA by taking into account information about other ranking positions, therefore identifies good compromise alternatives. This in particular makes sense when some extreme alternatives obtain the best ranking positions through some experts, but reach a very bad ranking position according to others.

We describe the preference structure among different experts that can be represented by a real-valued utility function $u(x_i, \lambda)$, which maps different alternatives $x_i$ to utility values

$$u_i(\lambda) = u(x_i, \lambda),$$

(9)

in terms of a weight vector $\lambda$ to quantify each specific preference among various decision results. Consider a more general environment in which neither input data nor weights are

exactly known. The uncertain or imprecise input data is represented by stochastic variables $\zeta_{il}$ with estimated joint probability distribution and density function $f(\zeta)$ in the space $X$, while the unknown or partially known preferences are represented by a weight distribution with density function $f(\lambda)$ in the set of feasible weights $\Lambda$ defined as

$$\Lambda = \left\{ \lambda \in \mathfrak{R}^p : \lambda \geq 0, \sum_l \lambda_l = 1 \right\}. \tag{10}$$

The set of feasible weights is therefore a $(p - 1)$ dimensional simplex. The aforementioned utility function is then employed to map stochastic input data and weight distributions into utility distributions $u(\zeta_i, \lambda)$.

Total loss of knowledge on weights is represented in "Bayesian" manner by a uniform weight distribution in $\Lambda$, which has density function

$$f(\lambda) = \frac{1}{vol(\Lambda)} = \frac{(p - 1)!}{\sqrt{p}}. \tag{11}$$

In SMAA, the set of favorable weights for each alternative $\Lambda_i(\zeta)$ is then defined as:

$$\Lambda_i(\zeta) = \{\lambda \in \Lambda : u(\zeta_i, \lambda) \geq u(\zeta_k, \lambda), \forall k\}. \tag{12}$$

The ranking position of each alternative is defined as an integer from the best (= 1) to the worst (= $m$), in terms of a ranking function:

$$rank(\zeta_i, \lambda) = 1 + \sum_k \phi(u(\zeta_k, \lambda) > u(\zeta_i, \lambda)), \tag{13}$$

in which $\varphi(ture) = 1$ and $\varphi(false) = 0$.

In SMAA-2, the set of favorable weights for $\Lambda_i^r(\zeta)$ is defined as:

$$\Lambda_i^r(\zeta) = \{\lambda \in \Lambda : rank(\zeta_i, \lambda) = r\}. \tag{14}$$

A weight $\lambda \in \Lambda_i^r(\zeta)$ assigns utilities for the alternatives in this manner so that alternative $x_i$ reaches ranking position $r$.

The rank acceptability index $b_i^r$ is thereby defined as the expected volume of the set of favorable weights, and regarded as a measure of the variety of different valuations granting alternative $x_i$ achieves ranking position $r$. Meanwhile, the rank acceptability index is calculated as a multidimensional integral over the input data distributions and the favorable rank weights by means of

$$b_i^r = \int_X f(\zeta) \int_{\Lambda_i^r(\zeta)} f(\lambda) d\lambda d\zeta. \tag{15}$$

The rank acceptabilities can be utilized directly in the evaluation of alternatives. For large-scale problems, we introduce an iterative process, in which the $\kappa$ best ranking positions ($\kappa$br) acceptabilities are analyzed at each iteration $\kappa$:

$$a_i^\kappa = \sum_{r=1}^\kappa b_i^\kappa. \tag{16}$$

The kbr acceptabilities $a_i^\kappa$ is a measure of the variety of different valuations that assign alternative $x_i$ any of the $\kappa$ best ranking positions.

The problem of comparing alternatives through rank acceptabilities motivates us to propose a complementary method that integrates the rank acceptabilities into holistic acceptability indices $a_i^h$ for each alternative as:

$$a_i^h = \sum_r \beta^r b_i^r, \tag{17}$$

where $\beta^r$ are surrogate weights. The basic requirements for surrogate weights are nonnegative, normalized and nonincreasing when rank increases, namely, $\beta^1 \geq \beta^2 \geq \cdots \geq \beta^m \geq 0$. The elicitation of surrogate weights have been extensively studied in literature [4, 10, 13].

## 3. Empirical study

### 3.1. FHI 2018

Data has been universally regarded as one of the most important resources in modern health care. The collection, sharing and analyzing of data can help identify disease earlier, make hospitals become faster organizations, and transform the patient experience. Value defined in health care are tracked, measured and improved though data. The FHI 2018 analyzes data and conducts interviews with leaders that are making value-based health care happen around the world, to produce practical insights that health care leaders can apply for accelerating their path towards that goal. The fist chapter of FHI 2018 outlines how the Value Measure tool can form the basis of a positive platform for change across the countries it surveys, and reports the value delivered by health systems of 16 countries, which are shown in Table 2 below. We observe that Germany performs best in Access, Singapore has the best performance in Satisfaction and Efficiency. The 16-country average Value Measure is 43.48, and Singapore has the highest Value Measure across the 16 countries surveyed.

### 3.2. Result and analysis

We take into account all possible individual preferences among Access, Satisfaction and Efficiency: ASE: access ⩾ satisfaction ⩾ efficiency, AES: access ⩾ efficiency ⩾ satisfaction, SAE: satisfaction ⩾ access ⩾ efficiency, SEA: satisfaction ⩾ efficiency ⩾ access, EAS: efficiency ⩾ access ⩾ satisfaction, and ESA: efficiency ⩾ satisfaction ⩾ access. By means of the closed-form

**Table 2. Value measure by country in the FHI 2018.**

| Country | Access | Satisfaction | Efficiency | Normalized Access | Normalized Satisfaction | Normalized Efficiency |
|---|---|---|---|---|---|---|
| Australia | 65.05 | 66.85 | 25.87 | 0.79 | 0.97 | 0.38 |
| Brazil | 36.99 | 21.08 | 22.06 | 0.37 | 0.00 | 0.28 |
| China | 31.50 | 44.63 | 38.19 | 0.29 | 0.50 | 0.69 |
| France | 67.45 | 63.77 | 21.33 | 0.83 | 0.90 | 0.26 |
| Germany | 78.72 | 53.30 | 20.77 | 1.00 | 0.68 | 0.25 |
| India | 12.23 | 59.67 | 28.02 | 0.00 | 0.82 | 0.43 |
| Italy | 53.11 | 44.97 | 27.24 | 0.61 | 0.51 | 0.41 |
| Netherlands | 63.57 | 60.86 | 22.35 | 0.77 | 0.84 | 0.29 |
| Russia | 63.58 | 31.75 | 27.38 | 0.77 | 0.23 | 0.42 |
| Saudi Arabia | 43.59 | 62.75 | 44.17 | 0.47 | 0.88 | 0.85 |
| Singapore | 45.46 | 68.27 | 50.11 | 0.50 | 1.00 | 1.00 |
| South Africa | 29.21 | 39.53 | 11.09 | 0.26 | 0.39 | 0.00 |
| Spain | 51.43 | 66.50 | 27.79 | 0.59 | 0.96 | 0.43 |
| Sweden | 62.14 | 61.05 | 21.11 | 0.75 | 0.85 | 0.26 |
| United Kingdom | 54.38 | 55.18 | 26.25 | 0.63 | 0.72 | 0.39 |
| United States | 55.15 | 45.46 | 12.23 | 0.65 | 0.52 | 0.03 |

**Table 3. Interval decision matrix.**

| Country | ASE | AES | SAE | SEA | EAS | ESA |
|---|---|---|---|---|---|---|
| Australia | [0.7144,0.8822] | [0.5866,0.7944] | [0.7144,0.9699] | [0.6743,0.9699] | [0.3788,0.7144] | [0.3788,0.7144] |
| Brazil | [0.1862,0.3724] | [0.2178,0.3724] | [0.0000,0.2178] | [0.0000,0.2178] | [0.2178,0.3268] | [0.1406,0.2811] |
| China | [0.2898,0.4945] | [0.2898,0.4945] | [0.3944,0.4900] | [0.4945,0.5968] | [0.4922,0.6945] | [0.4945,0.6945] |
| France | [0.6659,0.8676] | [0.5465,0.8305] | [0.6659,0.9046] | [0.5835,0.9046] | [0.2624,0.6659] | [0.2624,0.6659] |
| Germany | [0.6436,1.0000] | [0.6240,1.0000] | [0.6436,0.8414] | [0.4654,0.6828] | [0.2481,0.6436] | [0.2481,0.6436] |
| India | [0.0000,0.4172] | [0.0000,0.4172] | [0.4089,0.8178] | [0.4172,0.8178] | [0.2169,0.4339] | [0.4172,0.6258] |
| Italy | [0.5117,0.6148] | [0.5117,0.6148] | [0.5063,0.5605] | [0.4601,0.5117] | [0.4139,0.5144] | [0.4139,0.5117] |
| Netherlands | [0.6346,0.8076] | [0.5304,0.7721] | [0.6346,0.8430] | [0.5658,0.8430] | [0.2886,0.6436] | [0.2886,0.6436] |
| Russia | [0.4720,0.7723] | [0.4720,0.7723] | [0.2261,0.4992] | [0.2261,0.4720] | [0.4175,0.5949] | [0.3218,0.4720] |
| Saudi Arabia | [0.4716,0.7341] | [0.4716,0.7341] | [0.6773,0.8830] | [0.7341,0.8830] | [0.6597,0.8478] | [0.7341,0.8654] |
| Singapore | [0.4998,0.8333] | [0.4998,0.8333] | [0.7499,1.0000] | [0.8333,1.0000] | [0.7499,1.0000] | [0.8333,1.0000] |
| South Africa | [0.2154,0.3232] | [0.1277,0.2554] | [0.2154,0.3910] | [0.1955,0.3910] | [0.0000,0.2154] | [0.0000,0.2154] |
| Spain | [0.5896,0.7760] | [0.5088,0.6600] | [0.6600,0.9625] | [0.6600,0.9625] | [0.4280,0.6600] | [0.4280,0.6952] |
| Sweden | [0.6181,0.7988] | [0.5037,0.7506] | [0.6181,0.8470] | [0.5519,0.8470] | [0.2568,0.6181] | [0.2568,0.6181] |
| United Kingdom | [0.5817,0.6783] | [0.5112,0.6339] | [0.5817,0.7226] | [0.5556,0.7226] | [0.3885,0.5817] | [0.3885,0.5817] |
| United States | [0.3971,0.6455] | [0.3374,0.6455] | [0.3971,0.5811] | [0.2729,0.5166] | [0.0292,0.3971] | [0.0292,0.3971] |

solutions obtained in Section 2.1, the pessimistic and optimistic results are derived to formulate the following interval decision problem as Table 3. As for this stochastic decision problem, we follow [12] to consider both Gaussian and Uniform distributions to implement the SMAA-2. [14] develops a open-source implementation of SMAA methods in java, which can be downloaded at http://smaa.fi/jsmaa/.

### 3.3 Gaussian distribution

We consider that the interval-valued data satisfies the Gaussian distribution, the mean and variance are simulated as [12]:

$$\mu_{it} = \frac{v_{it}^p + v_{it}^o}{2}, \tag{18}$$

$$(\sigma^2)_{it} = \frac{v_{it}^o - v_{it}^p}{6}. \tag{19}$$

The rank acceptability indices are easily obtained and vividly illustrated in Table 4 and Fig 1 below. In addition, we use the rank-order centroid approach (ROC) to elicit surrogate weights for constructing the holistic acceptability indices: $\beta^r = \frac{1}{16}\sum_{t=r}^{16}\frac{1}{t}, r = 1, 2, \ldots, 16$. It is observed that the first rank support of Singapore is 90.36% of the possibility, while the last rank supports of Brazil and South Africa are 46.23% and 53.22%, respectively. This implies that Singapore is most likely to be ranked at the first, Brazil and South Africa have the similar probability to realize the last rank.

### 3.4. Uniform distribution

Again, we take into account the uniform distribution and apply the open-source decision supporting software to calculate the rank acceptability indices and show them in the following

**Table 4. Rank acceptability indices and HAI calculated under Gaussian distribution.**

| Country | $b^1$ | $b^2$ | $b^3$ | $b^4$ | $b^5$ | $b^6$ | $b^7$ | $b^8$ | $b^9$ | $b^{10}$ | $b^{11}$ | $b^{12}$ | $b^{13}$ | $b^{14}$ | $b^{15}$ | $b^{16}$ | HAI |
|---|---|---|---|---|---|---|---|---|---|---|---|---|---|---|---|---|---|
| Australia | 0.0521 | 0.3051 | 0.4513 | 0.1466 | 0.0342 | 0.0081 | 0.0016 | 0.0006 | 0.0004 | 0.0000 | 0.0000 | 0.0000 | 0.0000 | 0.0000 | 0.0000 | 0.0000 | 0.1271 |
| Brazil | 0.0000 | 0.0000 | 0.0000 | 0.0000 | 0.0000 | 0.0000 | 0.0000 | 0.0000 | 0.0000 | 0.0000 | 0.0000 | 0.0000 | 0.0001 | 0.0264 | 0.5112 | 0.4623 | 0.0063 |
| China | 0.0000 | 0.0000 | 0.0009 | 0.0056 | 0.0155 | 0.0177 | 0.0189 | 0.0262 | 0.0414 | 0.1837 | 0.2191 | 0.3475 | 0.1144 | 0.0091 | 0.0000 | 0.0000 | 0.0297 |
| France | 0.0027 | 0.0253 | 0.0965 | 0.2036 | 0.3265 | 0.1856 | 0.0931 | 0.0390 | 0.0143 | 0.0081 | 0.0029 | 0.0021 | 0.0003 | 0.0000 | 0.0000 | 0.0000 | 0.0829 |
| Germany | 0.0398 | 0.0645 | 0.0924 | 0.1285 | 0.1438 | 0.1666 | 0.1418 | 0.1194 | 0.0623 | 0.0266 | 0.0088 | 0.0051 | 0.0004 | 0.0000 | 0.0000 | 0.0000 | 0.0824 |
| India | 0.0000 | 0.0000 | 0.0000 | 0.0001 | 0.0001 | 0.0006 | 0.0017 | 0.0038 | 0.0062 | 0.0643 | 0.0729 | 0.1444 | 0.3188 | 0.3640 | 0.0178 | 0.0053 | 0.0184 |
| Italy | 0.0000 | 0.0000 | 0.0000 | 0.0000 | 0.0000 | 0.0000 | 0.0003 | 0.0014 | 0.0086 | 0.4697 | 0.4145 | 0.1054 | 0.0001 | 0.0000 | 0.0000 | 0.0000 | 0.0307 |
| Netherlands | 0.0000 | 0.0006 | 0.0054 | 0.0372 | 0.1289 | 0.2595 | 0.3194 | 0.1834 | 0.0414 | 0.0146 | 0.0057 | 0.0037 | 0.0002 | 0.0000 | 0.0000 | 0.0000 | 0.0626 |
| Russia | 0.0000 | 0.0000 | 0.0001 | 0.0003 | 0.0003 | 0.0023 | 0.0048 | 0.0123 | 0.0282 | 0.1455 | 0.2457 | 0.2715 | 0.2287 | 0.0602 | 0.0001 | 0.0000 | 0.0251 |
| Saudi Arabia | 0.0012 | 0.5359 | 0.1884 | 0.1022 | 0.0722 | 0.0413 | 0.0294 | 0.0265 | 0.0023 | 0.0006 | 0.0000 | 0.0000 | 0.0000 | 0.0000 | 0.0000 | 0.0000 | 0.1238 |
| Singapore | 0.9036 | 0.0522 | 0.0245 | 0.0113 | 0.0044 | 0.0018 | 0.0017 | 0.0003 | 0.0001 | 0.0001 | 0.0000 | 0.0000 | 0.0000 | 0.0000 | 0.0000 | 0.0000 | 0.2033 |
| South Africa | 0.0000 | 0.0000 | 0.0000 | 0.0000 | 0.0000 | 0.0000 | 0.0000 | 0.0000 | 0.0000 | 0.0000 | 0.0000 | 0.0000 | 0.0000 | 0.0017 | 0.4661 | 0.5322 | 0.0059 |
| Spain | 0.0006 | 0.0161 | 0.1385 | 0.3538 | 0.2136 | 0.1514 | 0.0842 | 0.0379 | 0.0032 | 0.0006 | 0.0001 | 0.0000 | 0.0000 | 0.0000 | 0.0000 | 0.0000 | 0.0876 |
| Sweden | 0.0000 | 0.0003 | 0.0020 | 0.0105 | 0.0571 | 0.1440 | 0.2448 | 0.3775 | 0.1036 | 0.0351 | 0.0131 | 0.0113 | 0.0007 | 0.0000 | 0.0000 | 0.0000 | 0.0548 |
| United Kingdom | 0.0000 | 0.0000 | 0.0000 | 0.0003 | 0.0034 | 0.0211 | 0.0583 | 0.1717 | 0.6879 | 0.0507 | 0.0056 | 0.0010 | 0.0000 | 0.0000 | 0.0000 | 0.0000 | 0.0440 |
| United States | 0.0000 | 0.0000 | 0.0000 | 0.0000 | 0.0000 | 0.0000 | 0.0000 | 0.0000 | 0.0001 | 0.0004 | 0.0116 | 0.1080 | 0.3363 | 0.5386 | 0.0048 | 0.0002 | 0.0154 |

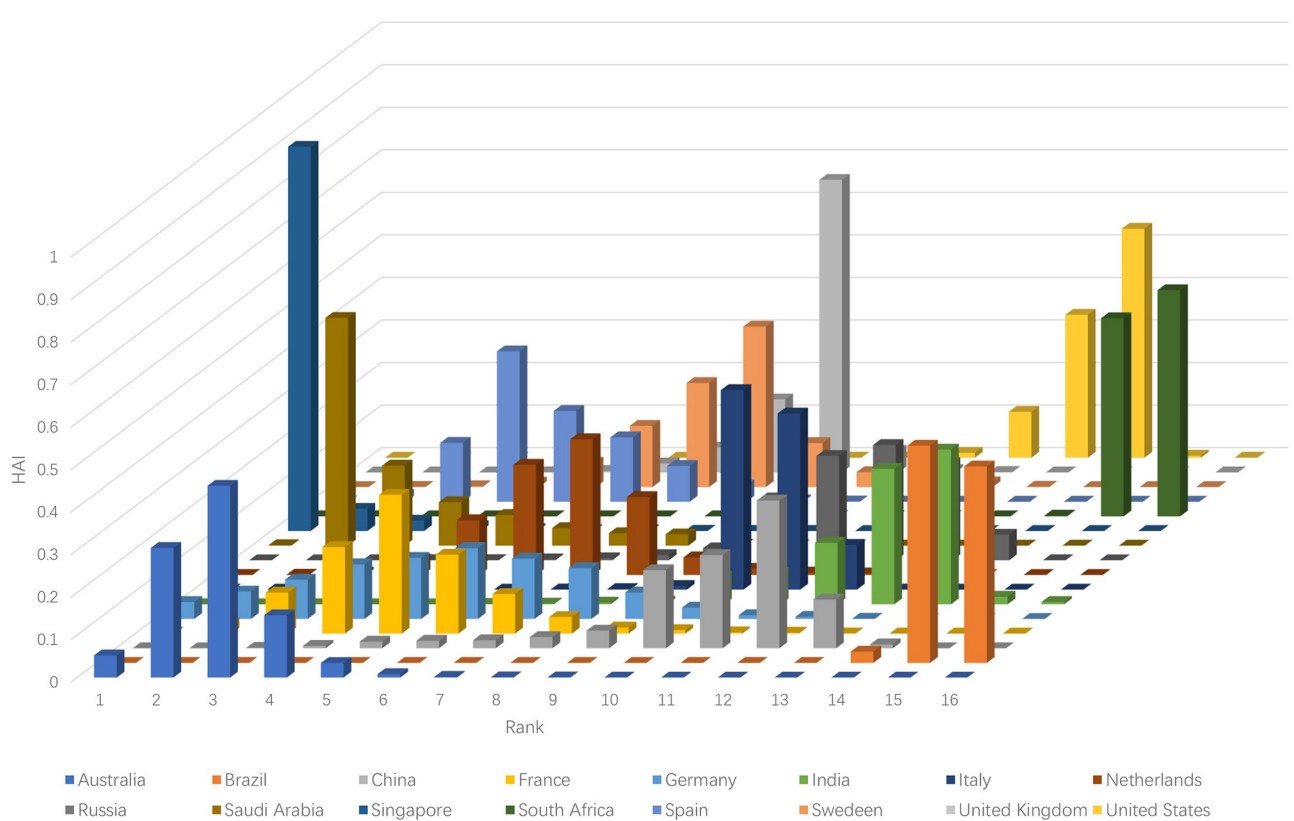

**Fig 1. Rank acceptability indices calculated under Gaussian distribution.**

**Table 5. Rank acceptability indices and HAI calculated under Uniform distribution.**

| Country | $b^1$ | $b^2$ | $b^3$ | $b^4$ | $b^5$ | $b^6$ | $b^7$ | $b^8$ | $b^9$ | $b^{10}$ | $b^{11}$ | $b^{12}$ | $b^{13}$ | $b^{14}$ | $b^{15}$ | $b^{16}$ | HAI |
|---|---|---|---|---|---|---|---|---|---|---|---|---|---|---|---|---|---|
| Australia | 0.0662 | 0.2542 | 0.3222 | 0.1874 | 0.0988 | 0.0432 | 0.0196 | 0.0060 | 0.0016 | 0.0004 | 0.0003 | 0.0001 | 0.0000 | 0.0000 | 0.0000 | 0.0000 | 0.1203 |
| Brazil | 0.0000 | 0.0000 | 0.0000 | 0.0000 | 0.0000 | 0.0000 | 0.0000 | 0.0000 | 0.0000 | 0.0000 | 0.0000 | 0.0000 | 0.0005 | 0.0484 | 0.4773 | 0.4738 | 0.0063 |
| China | 0.0000 | 0.0000 | 0.0023 | 0.0071 | 0.0153 | 0.0181 | 0.0276 | 0.0400 | 0.0662 | 0.1557 | 0.2263 | 0.2983 | 0.1235 | 0.0196 | 0.0000 | 0.0000 | 0.0306 |
| France | 0.0133 | 0.0587 | 0.1315 | 0.1633 | 0.1950 | 0.1590 | 0.1196 | 0.0818 | 0.0396 | 0.0181 | 0.0106 | 0.0077 | 0.0016 | 0.0002 | 0.0000 | 0.0000 | 0.0833 |
| Germany | 0.0421 | 0.0677 | 0.0944 | 0.1145 | 0.1256 | 0.1414 | 0.1414 | 0.1242 | 0.0770 | 0.0352 | 0.0185 | 0.0145 | 0.0033 | 0.0002 | 0.0000 | 0.0000 | 0.0807 |
| India | 0.0000 | 0.0000 | 0.0002 | 0.0000 | 0.0015 | 0.0025 | 0.0076 | 0.0133 | 0.0222 | 0.0721 | 0.0674 | 0.1448 | 0.3068 | 0.3248 | 0.0243 | 0.0125 | 0.0196 |
| Italy | 0.0000 | 0.0000 | 0.0000 | 0.0000 | 0.0000 | 0.0000 | 0.0032 | 0.0120 | 0.0391 | 0.4191 | 0.3994 | 0.1240 | 0.0032 | 0.0000 | 0.0000 | 0.0000 | 0.0310 |
| Netherlands | 0.0008 | 0.0072 | 0.0334 | 0.0959 | 0.1542 | 0.1963 | 0.2098 | 0.1707 | 0.0815 | 0.0256 | 0.0142 | 0.0081 | 0.0023 | 0.0000 | 0.0000 | 0.0000 | 0.0659 |
| Russia | 0.0000 | 0.0003 | 0.0001 | 0.0012 | 0.0031 | 0.0073 | 0.0141 | 0.0234 | 0.0415 | 0.1368 | 0.1893 | 0.2700 | 0.2304 | 0.0813 | 0.0012 | 0.0000 | 0.0258 |
| Saudi Arabia | 0.0173 | 0.4838 | 0.2047 | 0.1106 | 0.0682 | 0.0484 | 0.0338 | 0.0239 | 0.0072 | 0.0016 | 0.0005 | 0.0000 | 0.0000 | 0.0000 | 0.0000 | 0.0000 | 0.1228 |
| Singapore | 0.8546 | 0.0783 | 0.0303 | 0.0165 | 0.0082 | 0.0062 | 0.0034 | 0.0016 | 0.0007 | 0.0002 | 0.0000 | 0.0000 | 0.0000 | 0.0000 | 0.0000 | 0.0000 | 0.1988 |
| South Africa | 0.0000 | 0.0000 | 0.0000 | 0.0000 | 0.0000 | 0.0000 | 0.0000 | 0.0000 | 0.0000 | 0.0000 | 0.0000 | 0.0000 | 0.0000 | 0.0076 | 0.4808 | 0.5116 | 0.0060 |
| Spain | 0.0051 | 0.0471 | 0.1629 | 0.2433 | 0.2134 | 0.1590 | 0.1012 | 0.0531 | 0.0116 | 0.0030 | 0.0002 | 0.0001 | 0.0000 | 0.0000 | 0.0000 | 0.0000 | 0.0881 |
| Sweden | 0.0006 | 0.0027 | 0.0177 | 0.0579 | 0.1013 | 0.1624 | 0.2008 | 0.2165 | 0.1356 | 0.0476 | 0.0280 | 0.0225 | 0.0060 | 0.0004 | 0.0000 | 0.0000 | 0.0586 |
| United Kingdom | 0.0000 | 0.0000 | 0.0003 | 0.0023 | 0.0154 | 0.0562 | 0.1179 | 0.2333 | 0.4757 | 0.0806 | 0.0139 | 0.0044 | 0.0000 | 0.0000 | 0.0000 | 0.0000 | 0.0467 |
| United States | 0.0000 | 0.0000 | 0.0000 | 0.0000 | 0.0000 | 0.0000 | 0.0000 | 0.0002 | 0.0005 | 0.0040 | 0.0314 | 0.1055 | 0.3224 | 0.5175 | 0.0164 | 0.0021 | 0.0157 |

Table 5 and Fig 2. The aforementioned surrogate weights are employed to build the holistic acceptability indices. Similar to that of Gaussian distribution, the first rank support of Singapore is 85.46% of the possibility, while the last rank supports of Brazil and South Africa are 47.38% and 51.16%, respectively.

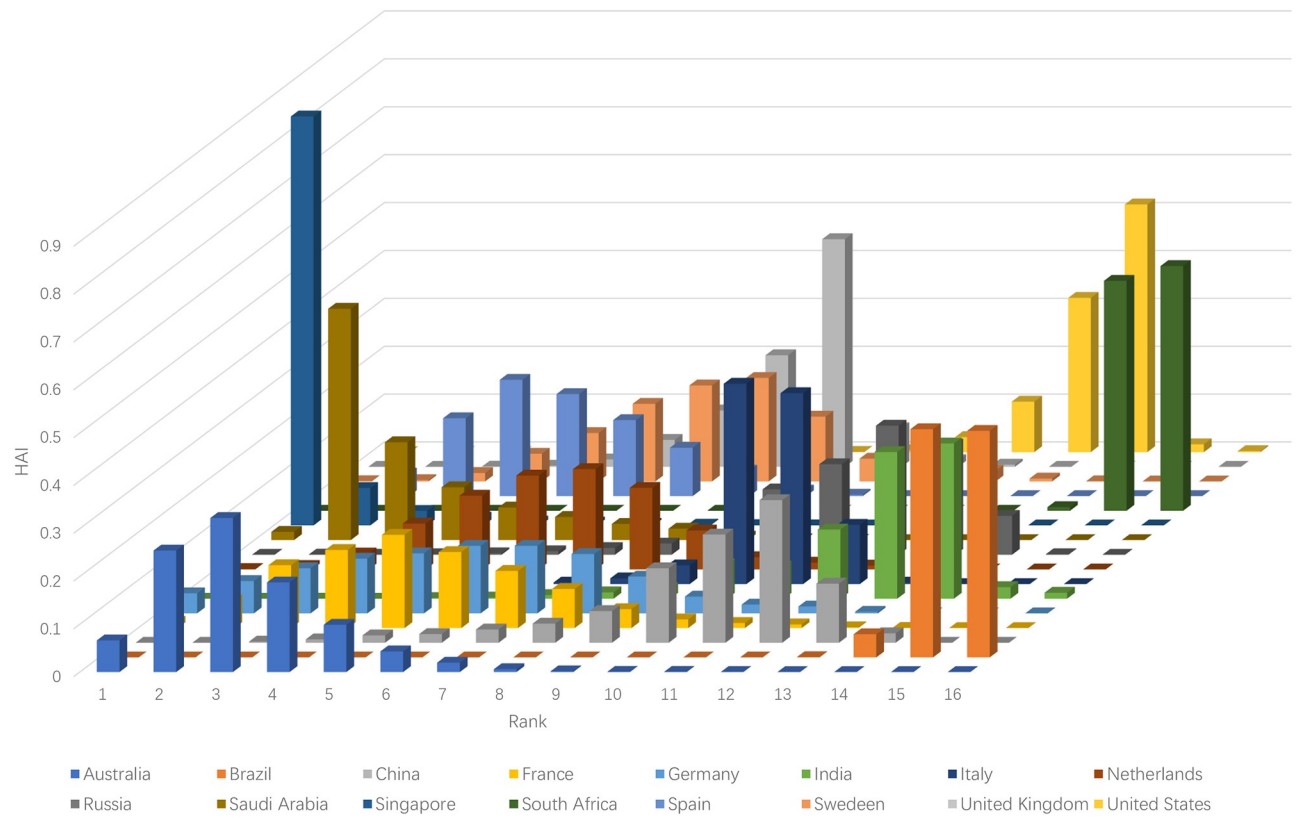

**Fig 2. Rank acceptability indices calculated under Uniform distribution.**

**Table 6. Rank comparisons.**

| Country | Value Measure | SMAA-2 with Gaussian distribution | SMAA-2 with Uniform distribution |
|---|---|---|---|
| Australia | 2 | 2 | 3 |
| Brazil | 15 | 15 | 15 |
| China | 12 | 11 | 11 |
| France | 4 | 5 | 5 |
| Germany | 3 | 6 | 6 |
| India | 14 | 13 | 13 |
| Italy | 10 | 10 | 10 |
| Netherlands | 6 | 7 | 7 |
| Russia | 11 | 12 | 12 |
| Saudi Arabia | 5 | 3 | 2 |
| Singapore | 1 | 1 | 1 |
| South Africa | 16 | 16 | 16 |
| Spain | 7 | 4 | 4 |
| Sweden | 8 | 8 | 8 |
| United Kingdom | 9 | 9 | 9 |
| United States | 13 | 14 | 14 |

In what follows, we use the holistic acceptability index under Gaussian and Uniform distributions as the revised metric of Value Measure, then compare these ranks with those according to the Value Measure, as shown in Table 6. It is evident that our method generates sufficiently robust rank among 16 countries. Only Australia and Saudi Arabia are ranked differently with slight difference. Meanwhile, the ranks of Brazil (15), Italy (10), Singapore (1), South Africa (16), Sweden (8) and United Kingdom (9) are significantly reliable because both our method and Value Measure produce the identical outcomes for them.

In addition, we make the full use of Spearman's rank correlation coefficient to verify the feasibility and rationality of the proposed method. In statistics, Spearman's rank correlation coefficient is a nonparametric measure between the rankings of two variables, and evaluates how well the relationship between two variables can be described using a monotonic function. The Spearman's rank correlation coefficient is capable of reflecting the conflict between ranking orders [15]. The more discordant the rankings of two variables, the smaller the Spearman's rank correlation coefficient [16]. The formula to compute Spearman's rank correlation coefficient is

$$\rho_s = 1 - \frac{6\sum_{i=1}^{m}(d_i)^2}{m(m^2 - 1)}, \quad (20)$$

where $d_i$ is the difference between the two ranks of each variable, and $m$ is the number of DMUs [17].

We calculate and compare the average Spearman's rank correlation coefficients in the following Table 7, which are capable of measuring the strength and direction of association

**Table 7. The superiority of our method.**

| Scenario | Access | Satisfaction | Efficiency | Average | Improvement |
|---|---|---|---|---|---|
| FHI 2018 | 0.6000 | 0.8000 | 0.2294 | 0.5431 | 0.00% |
| SMAA-2 with Uniform distribution | 0.3882 | 0.8706 | 0.4235 | 0.5608 | 3.25% |
| SMAA-2 with Gaussian distribution | 0.4147 | 0.8794 | 0.4029 | 0.5657 | 4.15% |

between obtained ranks and variables, and assessing the accuracy of models [18]. The Spearman's rank correlation coefficients between the Value Measure and Access, Satisfaction, Efficiency are computed as a benchmark for further analysis. Columns 2–4 report the Spearman rank correlation coefficients between the ranks obtained from our method and from Access, Satisfaction, Efficiency, respectively. Relative improvements are reported in the last column. Apparently, our method always outperforms the Value Measure, and the improvement from SMAA-2 with Gaussian and Uniform distributions are 4.15% and 3.25%, respectively.

According to the comparison of average Spearman's rank correlation coefficients, the proposed method outperforms the original Value Measure in terms of better associations between modified Value Measure and Access, Satisfaction, Efficiency. This indicates that countries can improve the levels of Value Measure in a precise manner.

## 4. Concluding remarks

The Future Health Index (FHI) 2018 measures and assesses the value presented in 16 health systems of developed and developing markets through proposing a broadly applicable composite indicator, namely, Value Measure, which is constructed in terms of the arithmetic average of Access, Satisfaction and Efficiency. However, the individual preferences among them remain largely unexplored in literature.

This paper proposes to apply the social choice theory and Stochastic Multicriteria Acceptability Analysis for group decision making (SMAA-2) for measuring the value of health systems, by means of re-constructing the Value Measure. Specifically, we begin with considering all possible individual preferences among Access, Satisfaction and Efficiency, which is mathematically represented by ranked weights of them; the pessimistic and optimistic outcomes under certain individual preference are derived in a closed-form manner, according to which an interval decision matrix is then formulated; the SMAA-2 is then applied to compute the holistic acceptability index and is considered as a revised Value Measure. An empirical study using the data of 16 countries is performed to demonstrate the usefulness of our method, in which both Gaussian and Uniform distributions have been taken into account. It is evident that our method is capable of generating sufficient robust and superior results to the Value Measure.

The applicability and feasibility of our method are in particular limited by two aspects of the data set: extreme values and number of metrics. Specifically, it is more meaningful to extensively investigate various individual preferences when the metric values are changed mildly among different metrics. Moreover, the application of our method could be more complicated when there exist more metrics to consider, since the importance orders would dramatically increase as the increase of the number of metrics. Therefore, the proposed method is applicable and feasible when the amount of metrics is considerably small, such as no more than four. For the scenario with over five metrics, future research should develop some statistical techniques, for example, principal component analysis, to select useful orders for implementation. In addition, future research should consider other statistical distributions (e.g., lognormal distribution, gamma distribution) of the stochastic parameters. A wide spectrum of methods should also be determined to select meaningful individual preferences for further analysis.

## Acknowledgments

The authors thank the editor and two anonymous reviewers for their helpful comments on our paper.

## Author Contributions

**Methodology:** Yelin Fu.

**Supervision:** Hai Shen.

**Writing – original draft:** Yelin Fu.

**Writing – review & editing:** Yubing Sui.

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
