## [Decision Letter · Decision Letter 0]

23 Dec 2019

PONE-D-19-26948

Using social choice theory and acceptability analysis to measure the value of health systems

PLOS ONE

Dear Dr. Fu,

Thank you for submitting your manuscript to PLOS ONE. After careful consideration, we feel that it has merit but does not fully meet PLOS ONE’s publication criteria as it currently stands. Therefore, we invite you to submit a revised version of the manuscript that addresses the points raised during the review process.

We would appreciate receiving your revised manuscript by Feb 04 2020 11:59PM. To enhance the reproducibility of your results, we recommend that if applicable you deposit your laboratory protocols in protocols.io, where a protocol can be assigned its own identifier (DOI) such that it can be cited independently in the future. For instructions see: http://journals.plos.org/plosone/s/submission-guidelines#loc-laboratory-protocols

We look forward to receiving your revised manuscript.

Kind regards,

Fausto Cavallaro, PhD

Academic Editor

PLOS ONE

https://www.philips.com/a-w/about/news/future-health-index/about.html

The text that needs to be addressed involves the last paragraph of the Introduction.

In your revision ensure you cite all your sources (including your own works), and quote or rephrase any duplicated text outside the methods section. Further consideration is dependent on these concerns being addressed.

a. Please provide an amended Funding Statement that declares *all* the funding or sources of support received during this specific study (whether external or internal to your organization) as detailed online in our guide for authors at http://journals.plos.org/plosone/s/submit-now

b. Please state what role the funders took in the study.  If any authors received a salary from any of your funders, please state which authors and which funder. If the funders had no role, please state: "The funders had no role in study design, data collection and analysis, decision to publish, or preparation of the manuscript."

Reviewers' comments:

Reviewer's Responses to Questions

**Comments to the Author**

1. Is the manuscript technically sound, and do the data support the conclusions?

Reviewer #1: Partly

Reviewer #2: Partly

2. Has the statistical analysis been performed appropriately and rigorously? 

Reviewer #1: I Don't Know

Reviewer #2: Yes

3. Have the authors made all data underlying the findings in their manuscript fully available?

Reviewer #1: No

Reviewer #2: Yes

4. Is the manuscript presented in an intelligible fashion and written in standard English?

Reviewer #1: No

Reviewer #2: Yes

5. Review Comments to the Author

Reviewer #1: The underlying concepts need to be defined more precisely. Geographic access, financial access, access to specialists appropriate for treating specific conditions? How is efficiency to be measured? What are the output and cost measures? In general it is not clear where the data come from? If these are guesstimates by experts, who are the experts? This lack of detail will greatly limit reader interest.

Reviewer #2: Positive comments:

The paper covers an interesting topic with improving a method to appropriately address the several factors that are relevant for the value of different health systems. Following an approach that takes rankings of these factors into account instead of weighting them equally seems a reasonable approach.

The overall logic of the paper is understandable. The problem with the existing methodology (computing the average overall) and the aimed contribution is pointed out. In general, the development of the methodology in the empirical part (section 2) is reasonable. Moreover, it is clearly a plus, that the author includes detailed information about the used software (e.g., for the SMAA-2) and the tested data set. This makes the method also applicable for researchers and practitioners in other contexts for similar problems.

The general demands from the journal are met, in that the study presents the results of original research, the methodology is overall described in a reasonable manner, conclusions seem to be supported by the data (at least partly, although there are some issue with this method outlined below) and standards of data and methodological transparency are obtained.

However, there is some potential to improve the paper, as pointed out below.

Methodological Development (Section 2)

The method of using ranked weights to improve the value measurement seems to be solid and bases on an established approach that is motivated from former research. However, some open questions and issues remain.

First, it is not clear, whether or how it is accounted for the several sub-metrics that build the metrics Satisfaction, Access and Efficiency. Are they still aggregated using the average method or are ranked weights used for them as well? If they are still aggregated, there would still be the issue of information loss within these metrics. Is there an approach or argument to deal with this issue?

Second, it remains unclear, how feasible this methodology is for a large set of indicators with many possible preference rankings. It appears to require a high amount of computational power and might get complex if many indicators are present. This limitation raises concerns about its applicability for a broader range of (more complex) problems. It would be helpful to add a statement of anticipated generalizability of the method.

Moreover, the argumentation and description of the methodological development (section 2) should be improved. Some decisions in the development are (apparently) made arbitrarily in that they are not justified convincingly. For the following points, the author should outline their reasons leading to these methodological decisions more clear:

1. Why is it necessary to normalize values, since the original metrics in the FHI are already normalized (on a scale from 0-100)?

2. Why is a uniform and a Gaussian distribution used?

3. The introduction of surrogate weights is very short and the mathematical calculation of them appears 2 pages after their introduction. This disrupts comprehension. It could also contribute the quick comprehension of the methodology, if the author could add a short explanation when introducing some new parameters (e.g., what k stands for (p. 7)).

Results of Empirical Analysis (Section 3)

The final comparison between the developed and the previous method to evaluate the value of health systems bases on Spearman’s Rank correlations. However, there are several issues with the presentation of this analysis:

1. It is not designated if the Spearman rank correlations are statistically significant.

2. The author provides a proportional improvement of the summed correlation coefficients, but does not provide a statistical index of whether this improvement (in %) is a significant improvement. The argumentation would be more convincing if an objective index would be added that proves a significant improvement in comparison to the previous method of averaging over metrics.

3. It is questionable whether the correlation with ranks based on the averaged metrics is appropriate, since it is argued in the paper before that averaging over indices has some methodological issues. Taking the ranks developed by the averaged metrics then as a comparison standard seems inconsistent with that former argumentation.

As such, it remains unclear, how to evaluate the improvement of the revised value measure in comparison to the former approach. For these reasons the soundness of conclusions based on data can be only confirmed partly.

Tables and Figures

The Tables and Figures are helpful and improve understanding in general.

For tables, however, information about the depicted measure could help to intuitive understanding them. Although mentioned in the text, this information should be included directly in the table or the title of the table or added as a footnote (see for example Table 4, 5, 7). In Table 7 significance of correlation coefficients should be added (see also comment above).

It would be nice, if information from Figure 1 and 2 could be joined allowing a direct comparison between both distributions.

Language

The author could improve wording and language use at some points to make her points really clear:

For example the sentence “Such an investigation sheds much-needed light on potential incentives and directions for academic, managerial and policy-related implications.” (p. 5) does not point out which implications are derived. The sentence “For the purpose of measuring the performance previously aggregated using the arithmetic average” (p. 5) is misleading, as it suggests that the former performance of the value measure is investigated, which is not the aim nor the outcome of the paper (instead, it compares the correlations of the sub-metrics with the old and the new values measure.

Furthermore, when describing the metrics of the FHI, the terms “sub-metric” and “metric” (introduced on p. 3) are used, but later on in the methodological part the term “sub-indices” refers to the metrics Access, Satisfaction and Efficiency, which can be confusing. It should be stated clearly once, which concepts from the application case in the empirical part refer to which concepts in the developed methodology.

Another issue is that some sentences in the introduction are nearly adopted literally, but are not marked as verbatim quotes. Concretely these are the following sentences:

“The challenges of delivering health care in many countries are receiving increasing attentions as costs continue to rise and evidence of uneven quality accumulates” (p. 2, first sentence, adopted from Porter, 2008), and “The Future Health Index (FHI)1 is a research-based platform designed by the Royal Philips to help determine the readiness of countries to address global health challenges and build sustainable...” (adopted from the FHI website). Both should be marked as verbatim quotes.

There is an (important) typo in Table 1 (p. 4). It is “healthy life expectancy” instead of “health life expectancy”.

Structure

In general the structure and the development of the paper makes sense. However, maybe the author could avoid repeating information and foster a fluent reading when slightly restructuring the paper.

In the current draft, FHI is described in detail already in the introduction, then the general methodological approach is developed and then the empirical section is introduced again with a description of the FHI. The detailed description of the FHI could also be positioned as an own section before the empirical section or included in the empirical section.

In the introduction, the objective and contribution of the study is mentioned relatively late (after introduction of the FHI). This lets the reader in the dark about the purpose of the study and disrupts a proper understanding from the very beginning.

Outline of Contribution

Although the paper apparently focuses on the development of a methodological approach, the motivation of the paper addresses several content-wise issues that are not really addressed later on in the paper. The author should become more clear what exactly the problem is they aim to address and to solve with their research (otherwise it appears like overstating the contribution of the paper).

As such, it remains unclear how applicable the findings are and which exact practical implications can be derived from them. It would contribute to the value of the paper from a broader perspective if the author would answer the following questions when shortly discussing the results:

1. How do the findings and the improvement of measurement contribute to achieving high value for various stakeholders?

2. Which (if any) content-wise and/or practical implications can be derived from the improved value measurement?

Concluding Remarks

Taking together, the paper covers an important topic with an immense societal relevance. It uses a convincing methodological approach that is theoretically reasoned to improve the value of health systems.

Major issues in the current draft and important implications for the further development of the paper will be to improve the scientific proof of empirically testing the improvement of the revised measure of the FHI in comparison to the former approach of averaging the conducted metrics. Moreover, the practical implications and methodological applicability of the results should be worked out more clearly.

Some concerns are also raised through the use of nearly verbatim quotes from other sources without designating them as such.

I wish the author good luck and look forward to the further development of the paper.

6. PLOS authors have the option to publish the peer review history of their article (what does this mean?). If published, this will include your full peer review and any attached files.

Reviewer #1: No

Reviewer #2: No

---

## [Decision Letter · Decision Letter 1]

28 Apr 2020

PONE-D-19-26948R1

Using social choice theory and acceptability analysis to measure the value of health systems

PLOS ONE

Dear Dr. Fu,

Thank you for submitting your manuscript to PLOS ONE. After careful consideration, we feel that it has merit but does not fully meet PLOS ONE’s publication criteria as it currently stands. Therefore, we invite you to submit a revised version of the manuscript that addresses the points raised during the review process.

We would appreciate receiving your revised manuscript by Jun 12 2020 11:59PM. To enhance the reproducibility of your results, we recommend that if applicable you deposit your laboratory protocols in protocols.io, where a protocol can be assigned its own identifier (DOI) such that it can be cited independently in the future. For instructions see: http://journals.plos.org/plosone/s/submission-guidelines#loc-laboratory-protocols

We look forward to receiving your revised manuscript.

Kind regards,

Fausto Cavallaro, PhD

Academic Editor

PLOS ONE

 Editor Comments: minor revision

Reviewers' comments:

Reviewer's Responses to Questions

**Comments to the Author**

1. If the authors have adequately addressed your comments raised in a previous round of review and you feel that this manuscript is now acceptable for publication, you may indicate that here to bypass the “Comments to the Author” section, enter your conflict of interest statement in the “Confidential to Editor” section, and submit your "Accept" recommendation.

Reviewer #2: (No Response)

Reviewer #3: All comments have been addressed

2. Is the manuscript technically sound, and do the data support the conclusions?

Reviewer #2: Partly

Reviewer #3: Yes

3. Has the statistical analysis been performed appropriately and rigorously? 

Reviewer #2: Yes

Reviewer #3: Yes

4. Have the authors made all data underlying the findings in their manuscript fully available?

Reviewer #2: Yes

Reviewer #3: Yes

5. Is the manuscript presented in an intelligible fashion and written in standard English?

Reviewer #2: No

Reviewer #3: Yes

6. Review Comments to the Author

Reviewer #2: One major issue of the previous submission is still not addressed sufficiently: The authors argue that averaging over sub-metrics to identify the value of health systems entails several problems (see p. 4). However, their whole analyses bases on averaged values (sub-metrics) which definitely weakens the soundness of their findings and the applicability of the developed method to the given data base. This substantial limitation of their findings due to the given data set needs to be mentioned, e.g., in the conclusion section. The authors should also address the question of how serious this concern might be in comparable data sets (i.e., how accessible are other than averaged results). So far it is not clear how feasible their method is to address the initially stated shortcoming of existing approaches to calculate value of health systems.

This issue becomes even more significant given the potential problems that their methodology conveys when faced with a larger number of metrics that could or should be used when calculating a value measure. Again, the authors should point out more clearly, when and where exactly their developed method is applicable.

The claim that the proposed method states a reasonable improvement to the former calculation method remains untested and is therefore still questionable. This is particularly relevant given the fact that the proposed methods requires a much higher level of effort and computational power and might not be applicable to similar problems with a higher number of included measures as basis for the value calculation.

Contribution

The contribution, goal and applicability of the paper’s finding should be pointed out more clearly. The added statement on page 5 “to measure the value of health systems” does not clarify the issue and is rather confusing. What exactly is meant with “sheds much-needed light on potential incentives and directions for academic, managerial and policy-related implications…”? This is not really a sentence at all. Please clarify. Note: This could also be a language issue.

Minor issues

Figures and tables

Again: Table 4, 5, and figures 1 and 2 do not contain information, which values are depicted. They are as such not intuitive. The main title or an added sub-title or footnote should directly provide information about what is depicted. (e.g. title of table 5 could easily be “Rank acceptability indices calculated under uniform distribution” or similar); the tables should be comprehensible on their own.

Reviewer #3: This communication presents an interesting novel method for measuring the value of health systems.

As per reviewers' suggestions, the authors have revised the paper successfully, but some grammatical errors are still present in the revised manuscript. Improve the errors.

7. PLOS authors have the option to publish the peer review history of their article (what does this mean?). If published, this will include your full peer review and any attached files.

Reviewer #2: No

Reviewer #3: No

---

## [Author Response · Author response to Decision Letter 1]

28 May 2020

Many thanks for your valuable comments on our manuscript. We make the revisions and highlight them in the revised version using RED. 

Reviewer #2: One major issue of the previous submission is still not addressed sufficiently: The authors argue that averaging over sub-metrics to identify the value of health systems entails several problems (see p. 4). However, their whole analyses based on averaged values (sub-metrics) which definitely weakens the soundness of their findings and the applicability of the developed method to the given data base. This substantial limitation of their findings due to the given data set needs to be mentioned, e.g., in the conclusion section. The authors should also address the question of how serious this concern might be in comparable data sets (i.e., how accessible are other than averaged results). So far it is not clear how feasible their method is to address the initially stated shortcoming of existing approaches to calculate value of health systems.

This issue becomes even more significant given the potential problems that their methodology conveys when faced with a larger number of metrics that could or should be used when calculating a value measure. Again, the authors should point out more clearly, when and where exactly their developed method is applicable.

Many thanks. The applicability and feasibility of our method are in particular limited by two aspects of the data set: extreme values and number of metrics. Specifically, it is more meaningful to extensively investigate various individual preferences when the metric values are changed mildly among different metrics. Moreover, the application of our method could be more complicated when there exist more metrics to consider, since the importance orders would dramatically increase as the increase of the number of metrics. Therefore, the proposed method is applicable and feasible when the amount of metrics is considerably small, such as no more than four. For the scenario with over five metrics, future research should develop some statistical techniques, for example, principal component analysis, to select useful orders for implementation.

The claim that the proposed method states a reasonable improvement to the former calculation method remains untested and is therefore still questionable. This is particularly relevant given the fact that the proposed method requires a much higher level of effort and computational power and might not be applicable to similar problems with a higher number of included measures as basis for the value calculation.

Many thanks. Spearman's rank correlation coefficients have been extensively utilized to justify the superiority of different ranking results, by means of computing and comparing the strength and direction of association between obtained ranks and variables. The results can be easily obtained by using Excel and do not need high level of effort and large computational power, even the number of metrics is large. 

Contribution

The contribution, goal and applicability of the paper’s finding should be pointed out more clearly. The added statement on page 5 “to measure the value of health systems” does not clarify the issue and is rather confusing. What exactly is meant with “sheds much-needed light on potential incentives and directions for academic, managerial and policy-related implications…”? This is not really a sentence at all. Please clarify. Note: This could also be a language issue.

Many thanks. We re-clarify the contribution on page 4: The contribution of this paper is the development of a new method to modify the Value Measure released by Philips for measuring the value of health systems, based upon the social choice theory and Stochastic Multicriteria Acceptability Analysis for group decision making (SMAA-2). The mentioned sentence on page 5 is rephrased as: This study proposes a new incentive and a feasible direction to measure the value of health systems in an appropriate manner, along with the provision of some academic, managerial and policy-related implications.

Minor issues

Figures and tables

Again: Table 4, 5, and figures 1 and 2 do not contain information, which values are depicted. They are as such not intuitive. The main title or an added sub-title or footnote should directly provide information about what is depicted. (e.g. title of table 5 could easily be “Rank acceptability indices calculated under uniform distribution” or similar); the tables should be comprehensible on their own.

Many thanks. The title of Tables 4&5, and Figures 1&2 are refined in the manuscript. 

Many thanks for your valuable comments on our manuscript. We make the revisions and highlight them in the revised version using BLUE.

Reviewer #3: This communication presents an interesting novel method for measuring the value of health systems.

As per reviewers' suggestions, the authors have revised the paper successfully, but some grammatical errors are still present in the revised manuscript. Improve the errors.

---

## [Decision Letter · Decision Letter 2]

18 Jun 2020

Using social choice theory and acceptability analysis to measure the value of health systems

PONE-D-19-26948R2

Dear Dr. Fu,

We’re pleased to inform you that your manuscript has been judged scientifically suitable for publication and will be formally accepted for publication once it meets all outstanding technical requirements.

Kind regards,

Fausto Cavallaro, PhD

Academic Editor

PLOS ONE

Additional Editor Comments:

Dear Authors,

The reviewer affirms that in the revised version, all comments have been addressed successfully. However, some language / grammar issues remain and should be edited before final publication. Another minor point would be the labeling of table 1; to support comprehension you could improve the title of table 1. It would be more appropriate if this is called “Factors underlying the Value Measure” or similar.

Reviewers' comments:

Reviewer's Responses to Questions

**Comments to the Author**

1. If the authors have adequately addressed your comments raised in a previous round of review and you feel that this manuscript is now acceptable for publication, you may indicate that here to bypass the “Comments to the Author” section, enter your conflict of interest statement in the “Confidential to Editor” section, and submit your "Accept" recommendation.

Reviewer #2: All comments have been addressed

2. Is the manuscript technically sound, and do the data support the conclusions?

Reviewer #2: Yes

3. Has the statistical analysis been performed appropriately and rigorously? 

Reviewer #2: Yes

4. Have the authors made all data underlying the findings in their manuscript fully available?

Reviewer #2: Yes

5. Is the manuscript presented in an intelligible fashion and written in standard English?

Reviewer #2: (No Response)

6. Review Comments to the Author

Reviewer #2: In the revised version, all comments have been addressed successfully. The revised contribution statement and limitation section contribute to the overall clarity and suggest some interesting potential directions for future research.

However, some language / grammar issues remain and should be edited before final publication. Another minor point would be the labeling of table 1; to support comprehension you could improve the title of table 1. It would be more appropriate if this is called “Factors underlying the Value Measure” or similar.

Otherwise, I congratulate the authors for the efforts they made for this paper.

7. PLOS authors have the option to publish the peer review history of their article (what does this mean?). If published, this will include your full peer review and any attached files.

Reviewer #2: No

---

## [Editor Report · Acceptance letter]

24 Jun 2020

PONE-D-19-26948R2 

Using social choice theory and acceptability analysis to measure the value of health systems 

Dear Dr. Fu:

I'm pleased to inform you that your manuscript has been deemed suitable for publication in PLOS ONE. Congratulations! Your manuscript is now with our production department. 

Kind regards, 

on behalf of

Professor Fausto Cavallaro 

Academic Editor

PLOS ONE